# Single-Nucleotide Variants in *PADI2* and *PADI4* and Ancestry Informative Markers in Interstitial Lung Disease and Rheumatoid Arthritis among a Mexican Mestizo Population

Karol J. Nava-Quiroz [1,2], Jorge Rojas-Serrano [3], Gloria Pérez-Rubio [1], Ivette Buendia-Roldan [4], Mayra Mejía [5], Juan Carlos Fernández-López [6], Espiridión Ramos-Martínez [7], Luis A. López-Flores [1], Alma D. Del Ángel-Pablo [1] and Ramcés Falfán-Valencia [1,*]

1 HLA Laboratory, Instituto Nacional de Enfermedades Respiratorias Ismael Cosío Villegas, Tlalpan, Mexico City 14080, Mexico; krolnava@hotmail.com (K.J.N.-Q.); glofos@yahoo.com.mx (G.P.-R.)
2 Programa de Doctorado en Ciencias Médicas, Odontológicas y de la Salud, Investigación Clínica Experimental en Salud, Bioquímica Clínica, Universidad Nacional Autónoma de México (UNAM), Mexico City 04510, Mexico
3 Rheumatology Clinic, Instituto Nacional de Enfermedades Respiratorias Ismael Cosío Villegas, Tlalpan, Mexico City 14080, Mexico
4 Translational Research Laboratory on Aging and Pulmonary Fibrosis, Instituto Nacional de Enfermedades Respiratorias Ismael Cosío Villegas, Tlalpan, Mexico City 14080, Mexico
5 Diffuse Interstitial Lung Disease Clinic, Instituto Nacional de Enfermedades Respiratorias Ismael Cosío Villegas, Tlalpan, Mexico City 14080, Mexico
6 Consorcio de Genómica Computacional, Instituto Nacional de Medicina Genómica (INMEGEN), Tlalpan, Mexico City 14610, Mexico
7 Experimental Medicine Research Unit, Facultad de Medicina, Universidad Nacional Autónoma de México, Mexico City 06720, Mexico
* Correspondence: rfalfanv@iner.gob.mx

**Abstract:** Rheumatoid arthritis (RA) is an autoimmune disease mainly characterized by joint inflammation. It presents extra-articular manifestations, with the lungs being one of the affected areas. Among these, damage to the pulmonary interstitium (Interstitial Lung Disease—ILD) has been linked to proteins involved in the inflammatory process and related to extracellular matrix deposition and lung fibrosis establishment. Peptidyl arginine deiminase enzymes (PAD), which carry out protein citrullination, play a role in this context. A genetic association analysis was conducted on genes encoding two PAD isoforms: PAD2 and PAD4. This analysis also included ancestry informative markers and protein level determination in samples from patients with RA, RA-associated ILD, and clinically healthy controls. Significant single nucleotide variants (SNV) and one haplotype were identified as susceptibility factors for RA-ILD development. Elevated levels of PAD4 were found in RA-ILD cases, while *PADI2* showed an association with RA susceptibility. This work presents data obtained from previously published research. Population variability has been noticed in genetic association studies. We present data for 14 SNVs that show geographical and genetic variation across the Mexican population, which provides highly informative content and greater intrapopulation genetic diversity. Further investigations in the field should be considered in addition to AIMs. The data presented in this study were analyzed in association with SNV genotypes in *PADI2* and *PADI4* to assess susceptibility to ILD in RA, as well as with changes in PAD2 and PAD4 protein levels according to carrier genotype, in addition to the use of covariates such as ancestry markers.

**Dataset:** DOI 10.3390/cells12182235 (published manuscript.). The datasets generated for this study can be found in ClinVar accessions SCV001422427–SCV001422434.

**Keywords:** PAD4/*PADI4*; PAD2/*PADI2*; interstitial lung disease; rheumatoid arthritis; RA-ILD; AIM

## 1. Summary

This work presents data obtained from previously published research [1]. Rheumatoid arthritis (RA) is an autoimmune disease that is primarily inflammatory, affecting the joints [2,3]. Extra-articular manifestations of the disease have been described, including damage to the lung interstitium (interstitial lung disease, ILD) [4].

Regarding factors associated with developing both diseases (RA and RA-ILD), smoking has been described as one of the main risk factors, in addition to genetic predisposition and occupational and environmental factors [5,6]. These exposures and predisposing factors alter the pulmonary environment, initially causing inflammation in both the lung and the joint [7].

In the inflammatory process, several interleukins, such as IL-6 [8], IL-1β [9], and TNF-α [10], have been proposed as possible biomarkers. One of the markers used with high sensitivity and specificity for RA is anti-citrullinated peptide antibodies (ACPA). However, in the presence of ILD, there is usually no change in positivity or levels to indicate a worse prognosis [11]. ACPA are antibodies that recognize citrullinated proteins; this process is carried out by peptidyl arginine deiminases (PAD) [12,13].

It has been linked to the involvement of proteins participating in the inflammatory process and associated with extracellular matrix deposition and fibrosis establishment in the lungs, such as PAD enzymes [14]. The most abundant isoforms in the lung are PAD2 and PAD4 [15,16].

Multiple SNVs in genes encoding PAD enzymes have been described in different populations, such as rs1748033 in *PADI4*, which has been associated with susceptibility to RA in Japanese [17,18], Korean [19], Indian [20,21], Iranian [22], and Netherlands [23] populations, as well as altered PAD4 protein levels, which correlate with the presence of anti-citrullinated peptide antibodies and levels of inflammatory proteins such as rheumatoid factor and C-reactive protein [11,24].

The prevalence of musculoskeletal "Ise'ses in Mexico has been reported to vary by geographic location [25], leading to an analysis of private markers within the Mexican population obtained by Silva et al. in 2009 [26]. Therefore, exposure to biomass burning smoke, smoking, PAD2 and PAD4 protein levels, SNVs in the genes encoding these enzymes, and ancestry markers were assessed in the Mexican population in groups of patients with interstitial lung disease, rheumatoid arthritis, and individuals without the disease.

## 2. Data Description

The data obtained in each analysis are presented and described in the tables and figures.

### 2.1. Study Groups

The included groups consisted of patients with rheumatoid arthritis (RA group), RA associated with interstitial lung disease (RA-ILD group), as well as a group of clinically healthy subjects (CHS group). These groups are displayed in the result tables.

### 2.2. Genotyping

Table 1 displays the analysis variable at the top of each column, including the gene, the SNV with their respective genotypes or alleles, and the patient or control group. The rows represent the frequency in percentage.

In Figure 1, diamond plots depict haplotype blocks identified in *PADI2* and *PADI4* within the Mexican mestizo population. The diamonds, connected by lines to form a block, represent a haplotype. At the top of the graph, you can see the seven SNVs, with each diamond symbolizing the linkage between these loci. Diamonds with a deeper shade of red indicate a higher degree of linkage disequilibrium, while lighter diamonds suggest a lower degree of linkage disequilibrium. The diamonds also display the linkage disequilibrium as 'r' (Pearson's Correlation Coefficient).

**Table 1.** Analysis of genotypes and alleles of SNV in *PADI2* and *PADI4*.

| Gene | Genotype or Allele | RA-ILD (*n* = 118) F% | RA (*n* = 133) F% | CHS (*n* = 616) F% |
|---|---|---|---|---|
| | rs2057094 | | | |
| | GG | 35.59 | 40.54 | 43.67 |
| | GA | 23.73 | 22.52 | 30.19 |
| | AA | 40.68 | 36.94 | 26.14 |
| | G | 47.46 | 51.8 | 58.77 |
| | A | 52.54 | 48.2 | 41.23 |
| | rs2076615 | | | |
| | AA | 46.09 | 42.73 | 30.86 |
| | AC | 50.43 | 53.64 | 63.86 |
| *PADI2* | CC | 03.48 | 03.64 | 05.28 |
| | A | 71.30 | 69.55 | 62.79 |
| | C | 28.70 | 30.45 | 37.21 |
| | rs1005753 | | | |
| | TT | 48.72 | 63.06 | 46.72 |
| | TG | 44.44 | 35.14 | 44.10 |
| | GG | 06.84 | 01.80 | 09.18 |
| | T | 70.94 | 80.63 | 68.77 |
| | G | 29.06 | 19.37 | 31.23 |
| | rs11203366 | | | |
| | GG | 21.55 | 35.96 | 30.50 |
| | GA | 52.59 | 50.00 | 47.00 |
| | AA | 25.86 | 14.04 | 20.90 |
| | G | 47.84 | 60.96 | 54.85 |
| | A | 52.16 | 39.04 | 45.14 |
| | rs11203367 | | | |
| | TT | 22.52 | 36.52 | 29.70 |
| | TC | 59.46 | 51.30 | 48.30 |
| | CC | 18.02 | 12.17 | 20.60 |
| | T | 52.25 | 62.17 | 54.60 |
| *PADI4* | C | 47.75 | 37.83 | 45.40 |
| | rs1748033 | | | |
| | CC | 16.38 | 14.91 | 25.10 |
| | CT | 50.00 | 51.57 | 49.80 |
| | TT | 33.62 | 33.33 | 23.80 |
| | C | 41.38 | 40.79 | 50.66 |
| | T | 58.62 | 59.21 | 49.34 |
| | rs874881 | | | |
| | CC | 23.28 | 44.62 | 24.70 |
| | CG | 59.48 | 42.31 | 51.50 |
| | GG | 17.24 | 13.08 | 22.90 |
| | C | 53.03 | 65.77 | 50.90 |
| | G | 46.95 | 34.23 | 49.10 |

F%: Frequency in Percentage; RA-ILD: rheumatoid arthritis associated to interstitial lung disease group; RA: rheumatoid arthritis group; CHS: clinically healthy subjects.

The haplotype analysis was conducted between all groups of patients with RA (with and without ILD) and the group of individuals without the diseases (Figure 1a), as well as within RA-ILD patients vs. RA patients (Figure 1b).

Table 2 displays the percentage frequencies of the haplotypes identified in *PADI4* (rs11203366-rs11203367-rs1748033-rs874881) within the patient populations of RA-ILD, RA and the group of individuals without the diseases (clinically healthy subjects-CHS), as shown in Figure 1a,b.

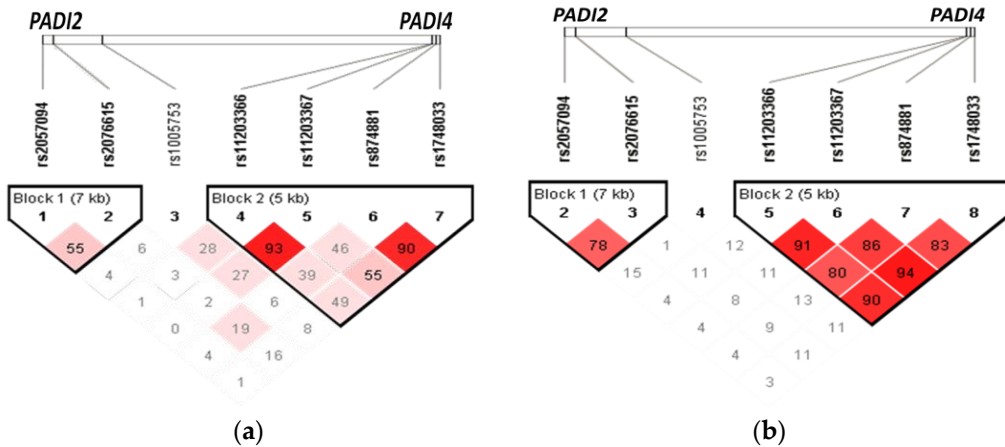

**Figure 1.** Haplotypes in *PADI2* and *PADI4*. (**a**) All Rheumatoid Arthritis patients vs. CHS; (**b**) RA-ILD vs. RA patients. The intensity of the red depends on the linkage disequilibrium; the more intense the red, the greater the binding disequilibrium.

**Table 2.** Analysis of genotypes and alleles of SNV in *PADI4*.

| *PADI4* | RA-ILD (*n* = 118, HF%) | RA (*n* = 133, HF%) | CHS (*n* = 616, HF%) |
|---|---|---|---|
| Haplotype | HF (%) | HF (%) | HF (%) |
| GTTC | 39.80 | 47.40 | 35.10 |
| ACCG | 37.70 | 32.20 | 30.80 |
| GTCG | 5.00 | 9.90 | 13.10 |
| ACTC | 7.20 | 2.90 | 10.60 |
| GTCC | 2.40 | 1.40 | 4.00 |
| ATTC | 2.30 | 2.50 | 1.10 |
| GCCG | 2.00 | 2.40 | 1.20 |

HF% = haplotype frequency in percentage.

### 2.3. AIMs

The genotyping of ancestry informative markers consists of 14 private SNVs between chromosomes 1 and 13. These markers exhibit differential minor allele frequencies (MAF) between the ZAP and CEU populations (taken as reference). Table 3 presents the results of the minor allele frequency in the reference populations, and at the end of the table, you can find the MAF for the study population (*n* = 867 individuals).

**Table 3.** AIMs polymorphisms used for the calculation of Eigenvectors.

| Chr | SNV | | ZAP (*n* = 60) | | | | CEU (*n* = 120) | | D | Study Group (*n* = 867) | | |
|---|---|---|---|---|---|---|---|---|---|---|---|---|
| No. | rs | A1 | A2 | MAF | A1 | A2 | MAF | Δ | A1 | A2 | MAF |
| 1 | rs4528122 | T | C | 0.067 | C | T | 0.142 | 0.792 | T | C | 0.338 |
| 1 | rs986690 | G | A | 0.017 | A | G | 0.25 | 0.733 | G | A | 0.347 |
| 4 | rs10516422 | G | A | 0.283 | G | A | 0.017 | 0.267 | G | A | 0.234 |
| 5 | rs10515716 | T | C | 0.267 | C | T | 0.208 | 0.525 | T | C | 0.432 |
| 6 | rs1878071 | A | C | 0.317 | C | A | 0.217 | 0.467 | A | C | 0.49 |
| 9 | rs4084051 | T | C | 0.25 | C | T | 0.175 | 0.575 | T | C | 0.483 |
| 9 | rs7853112 | C | A | 0.25 | A | C | 0.35 | 0.4 | C | A | 0.395 |
| 9 | rs10511491 | C | T | 0.25 | T | C | 0.391 | 0.358 | C | T | 0.358 |
| 9 | rs1039336 | A | G | 0.133 | G | A | 0.242 | 0.625 | G | A | 0.433 |
| 9 | rs10116714 | A | G | 0.183 | G | A | 0.05 | 0.767 | A | G | 0.449 |

**Table 3.** *Cont.*

| Chr | SNV | | ZAP (n = 60) | | | | CEU (n = 120) | | | D | | Study Group (n = 867) | |
|-----|-----|---|---|---|---|---|---|---|---|---|---|---|---|
| 9 | rs1980888 | G | A | 0.033 | A | G | 0.1 | 0.866 | G | A | 0.411 | | |
| 9 | rs4743556 | C | T | 0.172 | T | C | 0.167 | 0.661 | T | C | 0.486 | | |
| 12 | rs6487927 | C | T | 0.033 | C | T | 0.475 | 0.442 | C | T | 0.275 | | |
| 13 | rs2147155 | 0 | T | 0 | G | T | 0.5 | 0.5 | G | T | 0.15 | | |

ZAP: Zapotecs, CEU: Utah Residents with Northern and Western European Ancestry, MAF: Minor Allele Frequency, A: allele1 and 2. D: delta value obtained between MAF in CEU and MAF in ZAP populations.

The assessment of genetic distances between the study groups was carried out using the 14 SNVs presented in Table 3. To do this, we conducted a group comparison analysis, and the results of the FST index, along with their corresponding *p*-values, are shown in Table 4 for the comparisons between the respective row and column.

**Table 4.** Estimated pairwise comparison of $F_{ST}$ index in study groups.

| Study Group | RA-ILD | RA |
|-------------|--------|-----|
| RA | 0.32 (0.578) | - |
| CHS | 0.34 (0.853) | 0.24 (0.043) |

RA: Rheumatoid Arthritis; RA-ILD: RA associated with Interstitial lung disease; CHS: clinically healthy subjects. The $F_{ST}$ index (*p*-value) is shown.

### 2.4. Eigenvectors PCA

Figure 2 displays a Principal Component Analysis (PCA) where the CEU and ZAP populations are taken as reference groups. In this analysis, the control group consists of clinically healthy individuals from the CHS group, and the case group includes all patients from the RA and RA-ILD groups. The first two vectors are shown, collectively providing more than 85% of the information within the population.

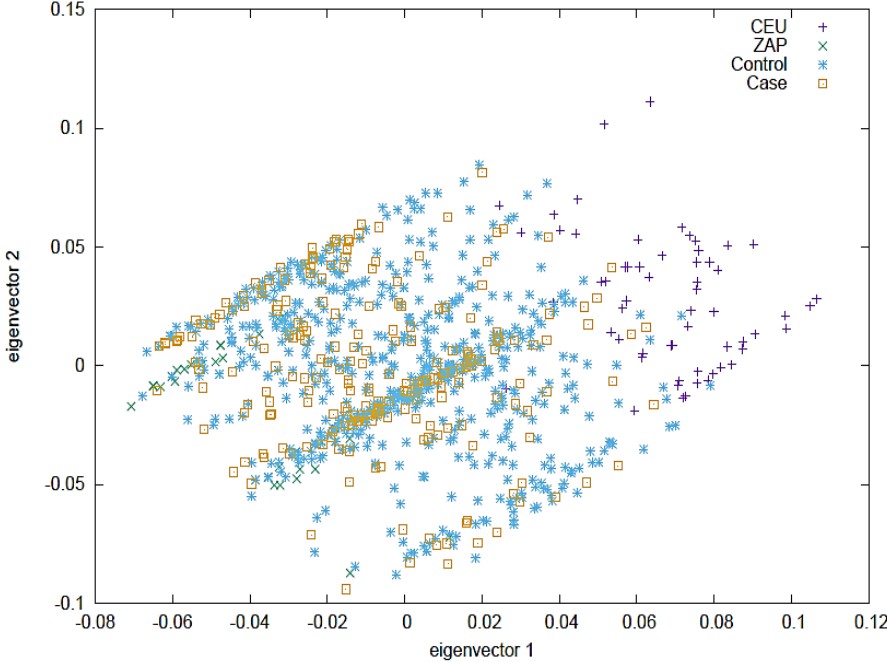

**Figure 2.** Principal component analysis, populations included in this study; the case group comprises patients with RA-ILD and with only RA (orange square), the control group (blue asterisk), and the reference populations CEU (plus purple symbol) and ZAP (green cross).

### 2.5. Exposures and Protein PAD2 and PAD4 Levels

In Figure 3a, the serum levels of PAD4 protein are displayed, and in Figure 3b, the PAD2 protein levels are shown for the three study groups. Each symbol (circle, triangle, and black square) represents an individual in whom the protein was measured.

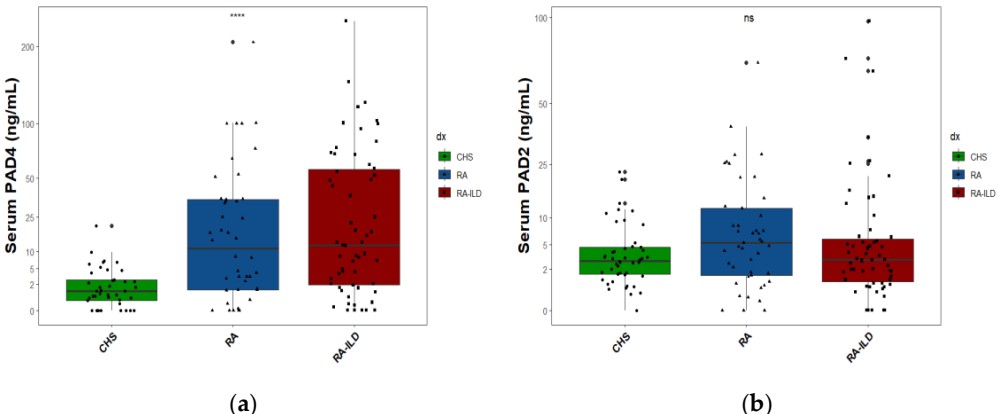

(**a**)　　　　　　　　　　　　(**b**)

**Figure 3.** Protein levels determined in serum within the three study groups: (**a**) PAD4 protein levels, (**b**) PAD2 protein levels. Each circle, triangle, or square represents an individual. CHS: clinically healthy individuals, RA: rheumatoid arthritis, RA-ILD: rheumatoid arthritis associated with interstitial lung disease. ****: Kruskal-Wallis, $p = 3.7 \times 10^{-6}$; ns: Kruskal-Wallis, $p = 0.18$.

Figure 4 displays the levels of PAD4 and PAD2 proteins according to exposure to smoking (a and c) or biomass (b and d), which are the primary environmental factors associated with the risk of both lung and joint diseases. In each figure, median values and interquartile ranges are depicted using box-and-whisker plots. The classification is based on smoking status or, in the case of exposure to biomass-burning smoke.

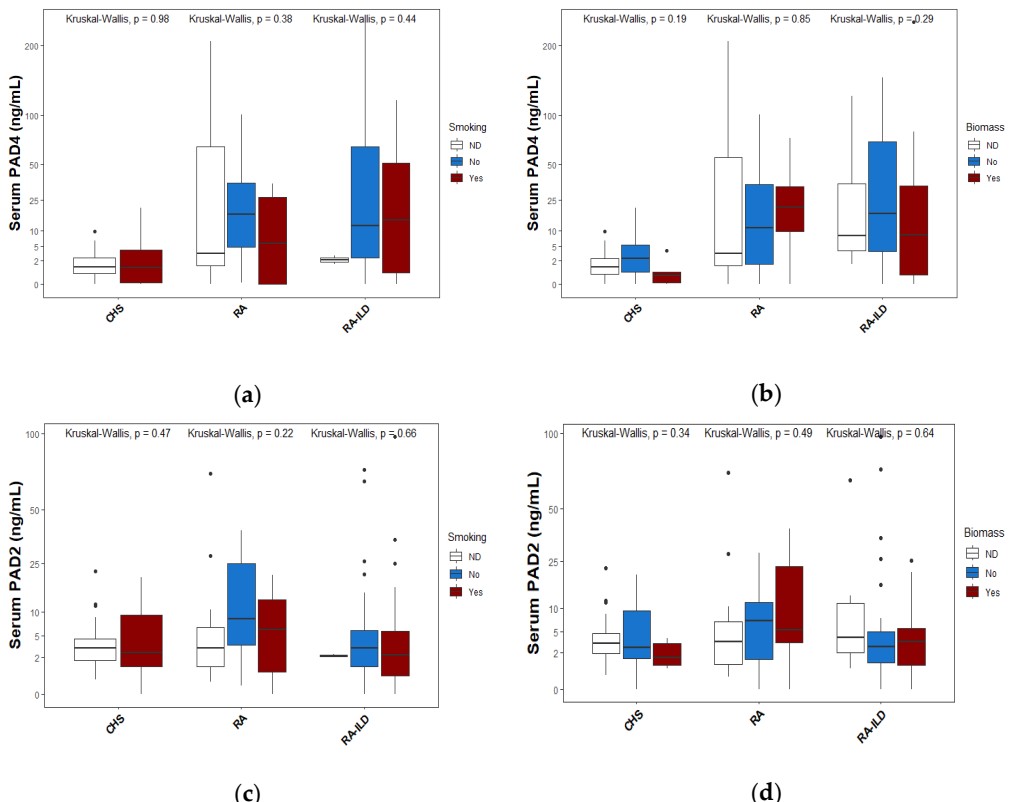

(**a**)　　　　　　　　　　　　(**b**)

(**c**)　　　　　　　　　　　　(**d**)

**Figure 4.** Protein levels determined in serum within the three study groups, classified by exposure to smoking or biomass: (**a**) PAD4 protein levels in individuals exposed to smoking. (**b**) PAD4 protein

levels in individuals exposed to biomass-burning smoke. (**c**) PAD2 protein levels in individuals classified by smoking status. (**d**) PAD2 levels based on biomass status. ND—Data not available. The dots outside the boxes and whiskers represent outliers.

*2.6. Proteins in BAL Samples*

Figure 5 presents the levels of PAD2 and PAD4 proteins, specifically measured in patients diagnosed with RA-ILD who underwent bronchoscopy for diagnostic purposes. Each square in the figure represents one sample from a patient.

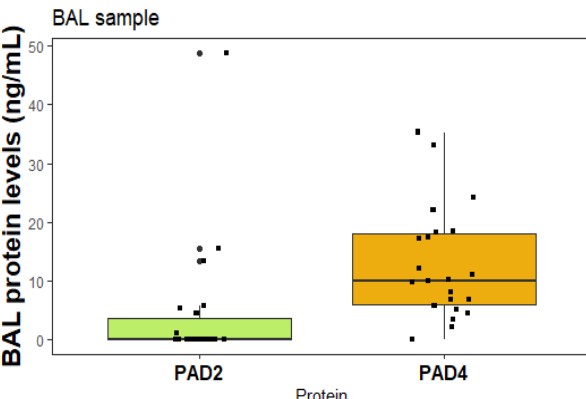

**Figure 5.** PAD2 and PAD4 protein levels determined in bronchioalveolar lavage (BAL) from 22 patients in the RA-ILD group.

## 3. Methods

The genetic data analysis was conducted using Plink v1.07 [27]. A total of 23 SNVs were assessed in 907 individuals, with the analysis divided into two parts: initially, 14 AIMs were examined independently, followed by an investigation of 8 SNVs, including four from *PADI2* and four from *PADI4*, for genetic association analysis.

A quality control process was implemented for all polymorphisms to ensure data quality. Individuals with more than 5% missing data, attributed to unsuccessful genotyping of SNVs, were excluded, resulting in the removal of 40 individuals. Additionally, a marker-level evaluation of the Hardy–Weinberg equilibrium was performed, leading to the exclusion of SNV rs2235926 in *PADI2* ($p = 8.9 \times 10^{-24}$).

Ancestry markers were analyzed in Plink v1.07, utilizing CEU and ZAP as reference populations, obtained from Phase 3 of the 1000 Genomes Project and the International HapMap Project (websites: http://browser.1000genomes.org and https://ftp.ncbi.nlm.nih.gov/hapmap, respectively; accessed 1 July 2021) [26,28]. The $F_{ST}$ index was assessed in R using the "Fine Pop" package, and distances among the three study groups were analyzed within a matrix, calculating the global $F_{ST}$ [29].

Protein levels were analyzed through non-parametric tests comparing the three groups. Medians and interquartile ranges were assessed using the Kruskal–Wallis test, and these statistics are presented in box-and-whisker plots for protein levels.

The graphs and statistical analyses of the protein and quantitative variables were carried out in the R environment (v.4.3.1), using the packages "ggplot2", "dplyr", "tidyverse", "ggpubr" and "stats" [29].

*Ethical Statement*

Approval for this study was obtained from the Research Institutional Committees for Research, Biosecurity, and Research Ethics at the Instituto Nacional de Enfermedades Respiratorias Ismael Cosío Villegas (INER) under the approval codes B20-15 and C08-15. All participants were granted their authorization and signed informed consent forms.

INER provided a document ensuring the safeguarding, including the adequate handling of personal data as sensitive and confidential information. The research adhered to the guidelines outlined in the 1975 Declaration of Helsinki.

**Author Contributions:** Conceptualization, K.J.N.-Q., J.R.-S., G.P.-R., I.B.-R. and R.F.-V.; Data curation, K.J.N.-Q., I.B.-R., M.M., J.C.F.-L. and E.R.-M.; Formal analysis, K.J.N.-Q., G.P.-R., M.M., J.C.F.-L., E.R.-M. and A.D.D.Á.-P.; Funding acquisition, R.F.-V.; Investigation, K.J.N.-Q., J.R.-S., G.P.-R., I.B.-R., M.M., J.C.F.-L., E.R.-M., A.D.D.Á.-P. and R.F.-V.; Methodology, K.J.N.-Q., G.P.-R., I.B.-R., J.C.F.-L., L.A.L.-F., A.D.D.Á.-P. and R.F.-V.; Project administration, R.F.-V.; Resources, J.R.-S., M.M., E.R.-M., L.A.L.-F. and R.F.-V.; Software, K.J.N.-Q., G.P.-R., J.C.F.-L., L.A.L.-F. and A.D.D.Á.-P.; Supervision, J.R.-S., M.M. and R.F.-V.; Validation, J.R.-S., G.P.-R., M.M., J.C.F.-L. and A.D.D.Á.-P.; Visualization, J.R.-S. and R.F.-V.; Writing—original draft, K.J.N.-Q., J.R.-S., G.P.-R., E.R.-M. and R.F.-V.; Writing—review and editing, K.J.N.-Q., G.P.-R. and R.F.-V. All authors have read and agreed to the published version of the manuscript.

**Funding:** This research received no external funding.

**Institutional Review Board Statement:** The research adhered to the principles of the Declaration of Helsinki and received approval from the Ethics Committee of the Instituto Nacional de Enfermedades Respiratorias "Ismael Cosío Villegas" (with protocol codes B20-15 and C08-15).

**Informed Consent Statement:** Consent was secured from all research participants, encompassing written consent and assurance of personal data confidentiality for publication purposes, both from the patients and healthy subjects who partook in the study.

**Data Availability Statement:** The datasets generated for this study can be found in ClinVar accessions SCV001422427–SCV001422434.

**Acknowledgments:** Programa de Maestría y Doctorado en Ciencias Médicas Odontológicas y de la Salud of the Universidad Nacional Autónoma de México (UNAM). Furthermore, the support provided by the Consejo Nacional de Humanidades, Ciencia y Tecnología (National Council of Science and Technology) (CONAHCyT), CVU: 690362.

**Conflicts of Interest:** All authors declare no conflicts of interest.

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
