# Peer review of "Single-Nucleotide Variants in PADI2 and PADI4 and Ancestry Informative Markers in Interstitial Lung Disease and Rheumatoid Arthritis among a Mexican Mestizo Population"

_data_

Round 1
Reviewer 1 Report
Comments and Suggestions for Authors
The present data descriptor type paper highlights the implications of single-nucleotide variants in PADI2 and PADI4 and ancestry 2 informative markers in interstitial lung disease and rheumatoid arthritis among a Mexican mestizo population. The topic is relevant, but some identified shortcomings in both content and form need to be addressed based on the specific recommendations below:
The abstract in the last part should include references to future perspectives based on the data presented and the results included in the former research.
The summary chapter is insufficiently presented to fully understand the purpose, the interrelationships between pathologies, the importance of biomarkers such as PAD, and the starting point of the research. I recommend consulting and inserting recent and relevant bibliographic sources for the topic: PMID: 34684377; PMID: 37193992; PMID: 33998910; PMID: 37031724.
Once rheumatoid arthritis is abbreviated in the main text (L43), only the abbreviated form will be used as long as it is in the main text (no need to re-abbreviate to L60).
The abbreviations of Table 1 should be explained at the end of the table as they are treated separately from the main text.
It is important to underline the statistical testing software with the version used.
Author Response
Dear Reviewer 1:
We have thoroughly reviewed each of the comments you provided. Your detailed insights and recommendations have been constructive in identifying areas where our article could be strengthened and clarified. Next, we outline how we have addressed your comments.
The present data descriptor type paper highlights the implications of single-nucleotide variants in PADI2 and PADI4 and ancestry 2 informative markers in interstitial lung disease and rheumatoid arthritis among a Mexican mestizo population. The topic is relevant, but some identified shortcomings in both content and form need to be addressed based on the specific recommendations below:
A: We would like to thank you for every one of the comments made on our manuscript; each of the changes made are marked in red or added to the text.
The abstract in the last part should include references to future perspectives based on the data presented and the results included in the former research.
R Thank you. Attending to your suggestion, we have added the following information:
Population variability has been noticed in genetic association studies. We show data for 14 SNVs that show a geographical and genetic variation between the population of México, which provides highly informative content and greater genetic diversity intrapopulation. Further investigations in the field should consider, in addition to AIMs, the genetic participation of PADI2 and PADI4-associated genotypes.
The summary chapter is insufficiently presented to fully understand the purpose, the interrelationships between pathologies, the importance of biomarkers such as PAD, and the starting point of the research. I recommend consulting and inserting recent and relevant bibliographic sources for the topic: PMID: 34684377; PMID: 37193992; PMID: 33998910; PMID: 37031724
A: The section has been restructured, and information has been added about the study of SNVs and its effects as a protein, shown in red in the text. We have added some articles that refer to us and information that helps us understand the issue. Thanks for your suggestion.
Once rheumatoid arthritis is abbreviated in the main text (L43), only the abbreviated form will be used as long as it is in the main text (no need to re-abbreviate to L60).
A: Line 60 contains the abbreviations for the groups in this study, so I have added the legend 'group' to each of the abbreviations to explain the label We used in the tables and figures.
The abbreviations of Table 1 should be explained at the end of the table as they are treated separately from the main text.
A: Line 60 contains the abbreviations for the groups in this study, so we added the legend 'group' to each of the abbreviations to explain the label I use in the tables and figures.
It is important to underline the statistical testing software with the version used.
A: The section on analysis using the software has been added to the last paragraph of the Methods section. “The graphs and statistical analysis of the protein and quantitative variables were carried out in the R environment (v.4.3.1), using the packages "ggplot2", "dplyr", "tidyverse", "ggpubr" and "stats" [9].”
Reviewer 2 Report
Comments and Suggestions for Authors Line 60-61: ‘diffuse’ is not included in the abbreviation ‘RA-ILD’
There are two abbriviations ‘RA-ILD’ and ‘ILD-RA. Please unify to ‘RA-ILD’.
Line 86: it is not clear how ‘the group of individuals without the diseases’ was abbreviated to ‘CHS’ in the main document.
Author Response
Dear Reviewer 2:
We have thoroughly reviewed each of the comments you provided. Your detailed insights and recommendations have been constructive in identifying areas where our article could be strengthened and clarified. Below, we outline how we have addressed your comments.
A: We would like to thank you for every one of the comments made on our manuscript; each of the changes made are marked in red or added to the text.
- Line 60-61: ‘diffuse’ is not included in the abbreviation ‘RA-ILD’
A: The word "diffuse" has been removed, leaving only the abbreviation and the previous description of the ILD.
- There are two abbreviations ‘RA-ILD’ and ‘ILD-RA. Please unify to ‘RA-ILD’.
A: changes were made to the abbreviation, as shown in tables and figures, to RA-ILD.
- Line 86: it is not clear how ‘the group of individuals without the diseases’ was abbreviated to ‘CHS’ in the main document.
A: The abbreviation includes a group of clinically healthy individuals (CHS). However, we refer to them in the text as individuals without the disease, as they do not have lung or joint disease. It was added in the description of the individuals on the same line (86).
Round 2
Reviewer 1 Report
Comments and Suggestions for Authors
The revision was poorly done. Please see my previous suggestions and proceed consequently, responding to each request, by improving your manuscript.
Author Response
Dear Reviewer:
The revision was poorly done. Please see my previous suggestions and proceed consequently, responding to each request, by improving your manuscript.
We sincerely thank you for your insightful comments on our manuscript. We have carefully reviewed and restructured the Abstract and Summary sections in response to your feedback. We trust that we have adequately addressed the remaining comments. Your careful review has proved invaluable, as your detailed insights and recommendations have been crucial in strengthening and clarifying critical aspects of our article. Below is a summary of how we have incorporated your feedback into our work.
The present data descriptor type paper highlights the implications of single-nucleotide variants in PADI2 and PADI4 and ancestry 2 informative markers in interstitial lung disease and rheumatoid arthritis among a Mexican mestizo population. The topic is relevant, but some identified shortcomings in both content and form need to be addressed based on the specific recommendations below:
A: We would like to thank you for each and every one of the comments made on our manuscript; each of the changes made are marked in red or added to the text.
The abstract in the last part should include references to future perspectives based on the data presented and the results included in the former research.
A: The citation of the RA-ILD association study has been included, with ancestry markers included as covariates. In addition, we have provided insights into the potential implications of the data in this manuscript.
The summary chapter is insufficiently presented to fully understand the purpose, the interrelationships between pathologies, the importance of biomarkers such as PAD, and the starting point of the research. I recommend consulting and inserting recent and relevant bibliographic sources for the topic: PMID: 34684377; PMID: 37193992; PMID: 33998910; PMID: 37031724
A: The section has been restructured to include details of the study of single nucleotide variations (SNVs) and their effect on protein, highlighted in red in the text. We've included one of the referenced articles and added additional information to enhance the understanding of the topic.
However, we have only included information related to our research topic, as rheumatoid arthritis (RA) is a complex disease influenced by many factors. While some of the proposed articles address other factors, such as the resolution of inflammation through prebiotics - a topic we find intriguing - we have chosen to focus on smoking, genetics, and protein modification by PAD enzymes. This approach may provide a unique perspective on RA-ILD.
We hope the current abstract meets the requirement of highlighting the central theme of PAD enzymes as a starting point to explore changes in protein levels and their impact on single nucleotide variations (SNVs) in RA-ILD. We believe that the remaining comments have been adequately addressed.
The remaining observations seem to us to have been made as appropriate:
Once rheumatoid arthritis is abbreviated in the main text (L43), only the abbreviated form will be used as long as it is in the main text (no need to re-abbreviate to L60).
A: Line 60 contains the abbreviations for the groups in this study, so I have added the legend 'group' to each of the abbreviations to explain the label I use in the tables and figures.
The abbreviations of Table 1 should be explained at the end of the table as they are treated separately from the main text.
A: Line 60 contains the abbreviations for the groups in this study, so I have added the legend 'group' to each of the abbreviations to explain the label I use in the tables and figures.
It is important to underline the statistical testing software with the version used.
A: The section on analysis using the software has been added to the last paragraph of the Methods section. “The graphs and statistical analysis of the protein and quantitative variables were carried out in the R environment (v.4.3.1), using the packages "ggplot2", "dplyr", "tidyverse", "ggpubr" and "stats" [9].”
Round 3
Reviewer 1 Report
Comments and Suggestions for Authors
The authors have significantly improved the manuscript based on the suggestions received.